# Theoretical-Experimental Analysis of the Performance of Geothermal Heat Pumps for Air Conditioning Greenhouses in Arid Zones

Jesús Octavio Rubalcaba Velasco [1], Alexis Acuña Ramírez [1,*], Jammin Abdi Quintal López [1], Abelardo Mercado Herrera [2], José Alejandro Suástegui Macías [1], Adolfo Heriberto Ruelas Puente [1], Fernando Lara Chávez [1], Pedro Francisco Rosales Escobedo [1] and José Armando Corona Sánchez [1]

1   Facultad de Ingeniería, Universidad Autónoma de Baja California (UABC), Mexicali 21280, Mexico
2   Programa Académico de Ingeniería Mecatrónica, Universidad Politécnica de Baja California (UPBC), Mexicali 21376, Mexico
*   Correspondence: alexis.acuna@uabc.edu.mx

**Abstract:** This study shows the results of a simulation tool using the TRNSyS 2017 simulator, validated with experimental data from a greenhouse in an arid zone in northwestern Mexico. Additionally, experimental data on the performance of geothermal heat pumps are shown during the year 2020 in heating and cooling mode. With this information, an average deviation of the simulator for the outlet fluid temperature of the geothermal heat exchanger (GHE) of 2.77% and an average deviation of the coefficient of performance in cooling mode (EER) of the geothermal heat pump (GHP) of 3.7% was obtained. In the experimental study, it was observed that in the last 2 weeks of July and the first 2 weeks of August, the subsoil is saturated, which causes a decrease in the thermal inertia of the GHE. During the experimental study, it was possible to determine that the flow indicated in the GHE to obtain the highest performance of the GHP system in greenhouses in arid zones corresponds to 1 GPM, obtaining an EER of 3.24.

**Keywords:** geothermal heat pump; geothermal heat exchanger; geothermal energy; greenhouse; TRNSyS simulation; renewable energies; sustainable technology

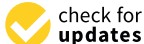



## 1. Introduction

Greenhouses are a technology used to produce food in areas where the weather and pollution can be negative factors during the harvest process. Greenhouses also reduce the water demand. The use of fossil fuels for greenhouse heating represents a high amount of greenhouse gas emissions into the environment. In this sense, the scientific community aware of this problem has made different proposals for sustainable technologies to meet the thermal and electrical demand of conventional greenhouses.

In 2018, Hassen et al. [1] carried out an analysis of the performance of a GHP connected to a novel conical GHE for heating a greenhouse in northern Tunisia. To carry out this research, the authors installed the GHP, the GHE, and the appropriate instrumentation to obtain experimental data. This study was made with the purpose of contributing to the resolution of the excessively cold temperatures during the night inside the greenhouse located in the north of Tunisia. Its greenhouse GHP system ensured an amount of heat equivalent to 692,208 kW, which corresponded to a temperature increase of 3 °C inside the greenhouse, with an optimal flow of 0.6 kg/s. In addition, the performance coefficients of the GHP and the system corresponded to 3.93 and 2.64, respectively.

In a different study, Ahmed O. [2] made a predictive model and an extensive numerical simulation of a microgrid for smart energy monitoring and management in a greenhouse. The author was motivated to achieve the net-zero energy goal in greenhouses by means of a predictive algorithm capable of capturing the complex interaction between microclimate

variables within the greenhouse and the development of plantations in any other greenhouse. The results showed that the control algorithm responded effectively to the control of heating, ventilation, and air conditioning operations, to monitor the ideal temperature of crops without using the electrical network, guaranteeing compliance with the objective of net-zero energy in greenhouses.

In another study, Mehdi M. et al. [3] designed and simulated a solar-geothermal system for heating a greenhouse. The authors carried out the simulation in the TRNSyS software to evaluate the performance of the system for a whole year, using refrigerant R-134A, R407C, and R410A as working fluids. The authors wanted to find a solution in which the soil recovers during the winter to prevent the COP of the system from decreasing due to high heat extraction from the soil when the thermal load is maximum. According to the results, the authors managed to increase the COP value by 0.6 compared to the system without subsoil preheating.

In New Zealand, Mariana de P. et al. [4] carried out an experimental analysis of the performance of six greenhouses heated with GHP. To obtain the experimental data, calorimeters were installed at the outlet of the vertical wells with concentric tubes to evaluate the heat gain of the GHE. The area where the greenhouses were installed was close to the native tribes of the region, and the study and expansion of the greenhouses directly benefitted the region. The results obtained from the heat gain of the geothermal well proved to be up to 4.5 times more than what was calculated, which implied that the greenhouses could be expanded and thus benefit the native community of the region.

In another relevant study, Mehmet E. et al. [5] developed an experimental study of different heat sources for greenhouse heating, such as biogas, solar energy, and geothermal energy. The installation and instrumentation of an experimental hybrid biogas greenhouse were carried out, including GHP with slinky-type GHEs and solar collectors for the supply of heat during the winter. The authors intended to demonstrate that heat sources such as biogas, solar energy, and geothermal energy can efficiently supply the energy needed for heating greenhouses in Eastern Turkey. It was concluded that the combination of biogas, solar energy, and geothermal energy has favorable results in the performance of greenhouses in Eastern Turkey and guarantees the autonomy of the system during the winter season.

Hassen B. et al. [6] conducted an experimental study to evaluate the performance of a novel conical GHE for cooling greenhouses under climatic conditions in Tunisia. A GHE array with a novel conical shape was designed, installed, and tested at a depth of 3 m. Performance analysis of the GHP and system showed a maximum COP of 4.25 and 3.25, respectively.

A numerical analysis of the long-term energy supply of a greenhouse using renewable energy sources was performed by Saeed M. et al. [7] a mathematical model of a microgrid and an interconnected microgrid were analyzed to evaluate the performance of the system and optimize it when considering the deviation of renewable energies. The authors wanted to find the perfect mix between the production of renewable energy, the number of storage units, and the capacity of the backup system that minimizes costs in the microgrid mode and maximizes remuneration in the interconnected mode. The results show that under certain climatic conditions in the Makran region, the combination of different renewable energies in the microgrid can cause an unjustified technical-economic proposal, that is, that the investment and maintenance costs could make the system unsustainable.

Alexandros S. et al. [8] developed a numerical analysis of a greenhouse that used photovoltaic energy, hydrogen, and geothermal energy for heating during the winter season. The mathematical model analyzed was of a photovoltaic array connected to an electrolyzer that, during the day, generated hydrogen, which was stored in a tank, and during the night, that hydrogen was consumed by a fuel cell to generate electricity to supply energy to a GHP with vertical GHEs for heating a tunnel greenhouse. The overall efficiency of the greenhouse heating system with renewable energy was 11%. Additionally, Issam M. et al. [9] presented a numerical-experimental analysis of a greenhouse air conditioning system that uses

groundwater in direct-indirect evaporative cooling equipment in the desert climate of Baghdad. The authors proposed a mathematical model that describes the thermal load of refrigeration in the greenhouse and the instrumentation in the greenhouse to obtain experimental data. It was possible to increase the efficiency of the direct or indirect evaporative cooling system up to 108% of the efficiency of the conventional system by using subsoil water as the cooling fluid. A different strategy to control the climate with an evaporation system with natural ventilation connected to a brackish water well was presented by Hacene M. et al. [10] In turn, this system allowed desalination of water for irrigation use in the greenhouse. The regions of the Middle East and North Africa have very little drinking water, but they do have brackish water and sources of geothermal energy. For this reason, the authors were motivated to propose a seawater desalination system and greenhouse air conditioning. A similar study presented by Youngguk S. et al. [11] analyzed a GHP greenhouse air-conditioning system in areas near highways in South Korea. The authors proposed a mathematical model that described the performance of GHP under climatic conditions in South Korea. Similarly, an experimental greenhouse was installed to validate the mathematical model. This study was carried out with the intention of being able to effectively predict the return on investment of greenhouses heated with GHP and compare them with conventional heating systems. This study showed that the experimental COP of the system varied between 3.4 and 3.6, compared to a design COP value of 3.7. Erdem C. et al. [12] presented a state-of-the-art focus on energy-saving strategies and air conditioning in greenhouses. The study focused on different technologies for photovoltaic modules, solar collectors, thermal energy storage systems, GHPs, lighting, and insolation in greenhouses. The intention of the authors was to show a broad overview of cutting-edge technology that improves greenhouse performance and reduces greenhouse gas emissions, as well as energy consumption and operating costs. The study showed that, in general, the use of greenhouse air conditioning strategies with renewable energies reduces the return on investment between 4 and 8 years, depending on the technology used.

In summary, the lines of research for greenhouse air conditioning using renewable energies focus mainly on geothermal energy and solar energy. Mexico is a world leader in the use of geothermal energy for electricity generation; however, this contrasts with the direct use of geothermal heat, which has received less attention. This is despite having large resources of medium and low enthalpy distributed throughout the country. However, applications of GHPs in tropical or semi-tropical climates are few. Within Mexico, only two industrial-type applications are known: balneology, in most regions of the country, and a district heating system for offices, workshops, and laboratories in the Los Azufres geothermal field (Romo J. et al. [13]). Approximately 73.7% of Mexico's area is classified as arid or semi-arid zones, which represents a potential for developing applications of this technology suitable for the particular conditions of the country. In particular, the performance of slinky-type systems should be characterized since, due to the soil conditions in these regions, it is easier in technological and economic terms to make trenches than to drill. Nowadays, there is no simulation tool with the specific parameters of the country's regions since climatic conditions can easily reach 50 °C and remain in that temperature band for a long period of the day, especially in extreme climate zones. Climatic conditions can also reach very low temperatures in winter. An area of research opportunity was found for simulation tools that allow the performance of GHP systems with slinky-type GHE to be predicted. The purpose of this study is to obtain and validate, through experimentation, a simulation tool using the TRNSyS software of a greenhouse air conditioning system in arid zones in northwestern Mexico. This tool allows forecasting the performance of GHE systems under very high and low-temperature climatic conditions. Additionally, the performance of the geothermal air conditioning system will be analyzed with the intention of evaluating and determining if the installed capacity in the experimental greenhouse meets the thermal demand.

## 2. Materials and Methods

The present study shows the results obtained through the simulation of GHE and its comparison with the results obtained experimentally on-site, as well as the materials and instrumentation used. As a result of the research, a validated GHE simulation tool is obtained in the TRNSYS software, which allows evaluating the performance of a horizontal slinky-type GHE and knowing the deviation between data taken experimentally on-site and the simulated values.

### 2.1. Materials and Instrumentation

The materials and instrumentation used during the experimental tests in the greenhouse are shown below. Figures 1 and 2 show the analog instrumentation that was used for temperature, pressure, and flow measurements. Table 1 shows the instrumentation used, the measurement ranges, and the measurement deviation.

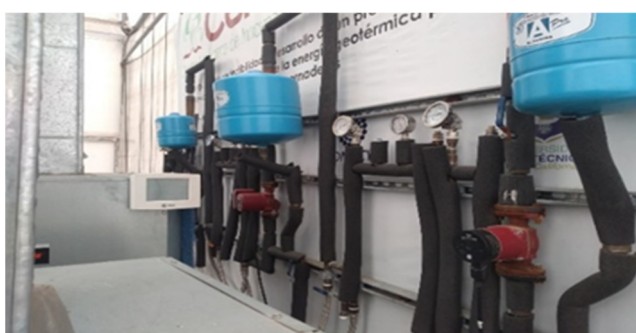

**Figure 1.** Geothermal heat pump hydraulic system installation in the greenhouse.

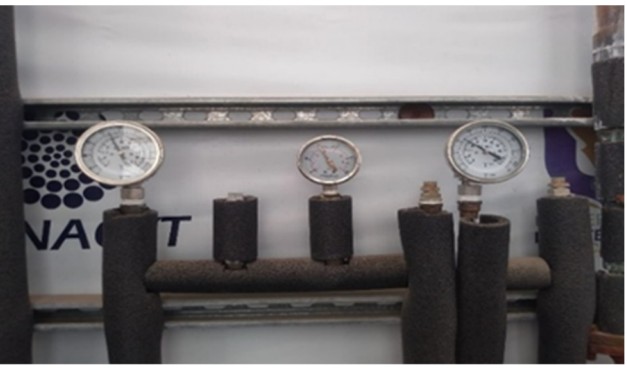

**Figure 2.** Analog instrumentation in the geothermal heat pump analysis was installed.

**Table 1.** Instrumentations details of the sensor used to analyze the geothermal heat pump's performance.

| Instrumentation | Measurement Ranges | Deviation |
|---|---|---|
| Analog Thermometer Analog | 0–100 °C | +/−1.6 of range of measurement |
| Manometer Analog Flowmeter | 0–700 kPa | +/−0.01 |
| Analog | 0–75 LPM | +/−0.05 |
| Ammeter | 0–100 A | +/−0.01 |

Figure 1 shows the recirculation pumps and the hydropneumatic tanks that allow each of the hydraulic systems to be kept pressurized.

The material used in the slinky-type GHEs is high-density polyethylene (HDPE), which is widely used in geothermal applications. Figure 3 shows the GHE in position before being buried. Each GHE has a linear length of pipe of 244 m per cooling ton of refrigeration. The general dimensions of the GHE are 30 m long and 1 m wide. The nominal diameter of the pipe used for its construction is 19 mm.

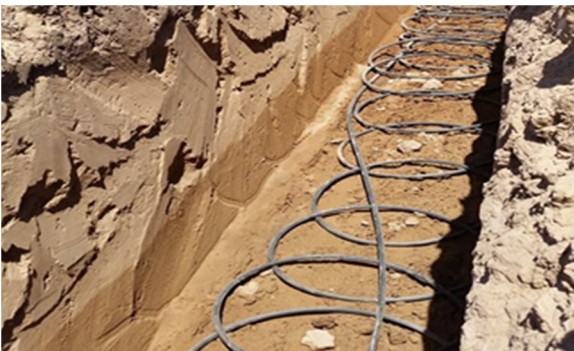

**Figure 3.** Slinky-type geothermal heat exchanger installation used in the GHP to assist the greenhouse.

### 2.2. Methods

The development of this study was carried out in the city of Mexicali, Baja California, which is located in northwestern Mexico, with the southwestern border of the United States of America at the geographic coordinates 32°43″ latitude and 115°56″ longitude. This city has an arid-dry climate and very little or no precipitation. A hot, dry climate is predominant in this city. The maximum temperature occurs in the month of July, when temperatures above 47 °C are reached, and the minimum temperature is reached in the month of January, when temperatures below 5 °C have been reported. Figure 4 shows the maximum, average, and minimum local temperature of Mexicali, México.

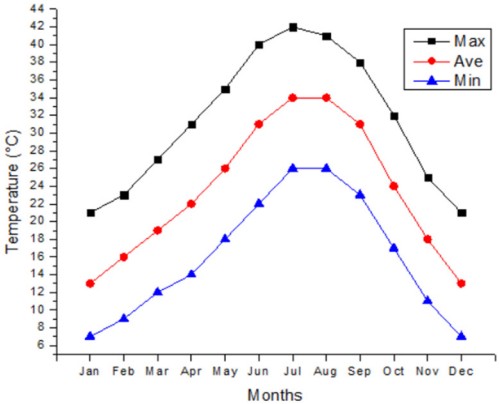

**Figure 4.** Maximum, average, and minimum local temperature of Mexicali during the season of harvest and operation of the GHP.

The experimental tests were carried out at the facilities of the Polytechnic University of Baja California (UPBC). This university has an experimental multi-tunnel type greenhouse heated by GHP with an area of 500 m$^2$. The greenhouse has a highly durable thermal polyethylene cover with a thickness of 200 μm. The experimental analysis consisted of obtaining data on the performance of the GHE with 3 different flow rates while the GHP was in cold mode, that is, refrigeration mode. Among the experimental data are the inlet and outlet temperatures of the GHE, the electrical consumption of the compressor, and the electrical consumption of the recirculation pump.

The mathematical model that defines the performance of slinky-type GHEs is usually complex to solve or simulate. For this reason, the intention is to perform a simulator of a linear GHE, which is relatively simpler to simulate and validate and compare with the experimental data obtained from a slinky-type GHE to estimate the degree of deviation and evaluate if it is possible to use this simulator in geothermal installations.

Simulator Considerations

The simulator and its validation were developed in the TRNSyS simulation software, using as a reference the information published by the authors Incropera et al. [14] in their book Heat and Mass Transfer, which is widely used in the area of engineering and mechanical design.

In the validation of the simulator, the same conditions of climate (winter), soil (West Lafayette, Indiana), and GHE material (HDPE) used by the authors were programmed.

During the simulation, the following parameters were used: inner tube diameter D = 25 mm; mass flows $\dot{m}_1 = 0.015$ kg/s, $\dot{m}_2 = 0.030$ kg/s and $\dot{m}_3 = 0.045$ kg/s; a fluid inlet temperature $T_{in} = 0\,°C$; and a soil thermal conductivity $k_{soil} = 0.47$ W/(m·K).

Figure 5 shows the scheme used in the simulation in TRNSYS. Each Type 952 represents a linear GHE with a different length in the range of (10 m $\leq$ L $\leq$ 50 m) according to the reference bibliographic information that was made for the validation of the simulator.

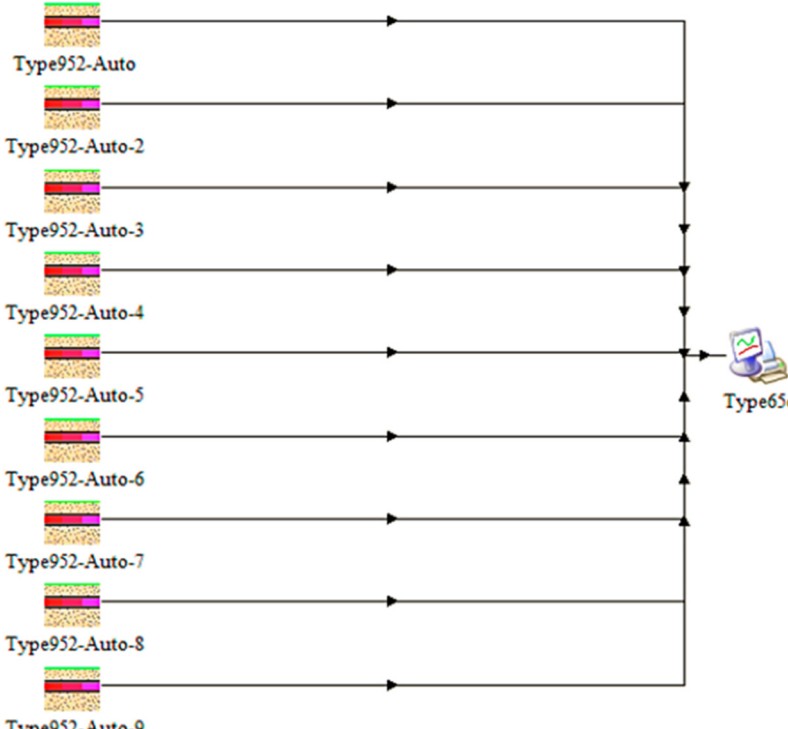

**Figure 5.** The scheme of the geothermal heat exchanger simulator builds on the TRNSyS 2017 software.

Once the simulator was validated, a simulation was carried out considering the real conditions of the GHE installation on site. The GHP configuration consisted of an installation of two GHPs in parallel, joined by a pair of flow distribution accessories, both for fluid inlet and outlet. In the same way, in the simulation, the flow rates and inlet temperatures to the flow distribution accessories measured experimentally were considered. The objective of this simulation was to compare the simulated and experimental outlet temperatures in order to obtain the deviation in the simulated data. In this way, the necessary data were obtained to calculate the theoretical and experimental performance of the GHP. Figure 6 shows the configuration of the TRNSyS 2017 simulator used in this study.

The tests on the GHPs were carried out at the UPBC facilities in its experimental greenhouse heated with geothermal energy. Figure 7 shows the GHP used for the experimentation, in the following order: 3, 5, and 2 tons of refrigeration, from left to right.

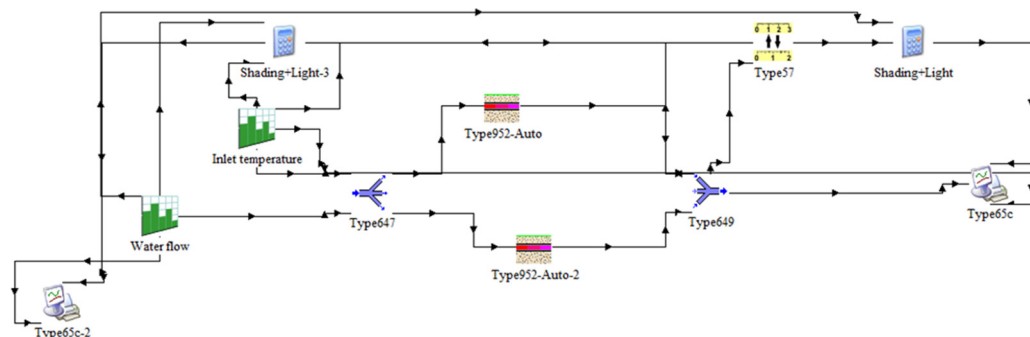

**Figure 6.** Simulator configuration of the geothermal heat pump to study the performance with TRNSyS 2017 under the weather conditions of the site.

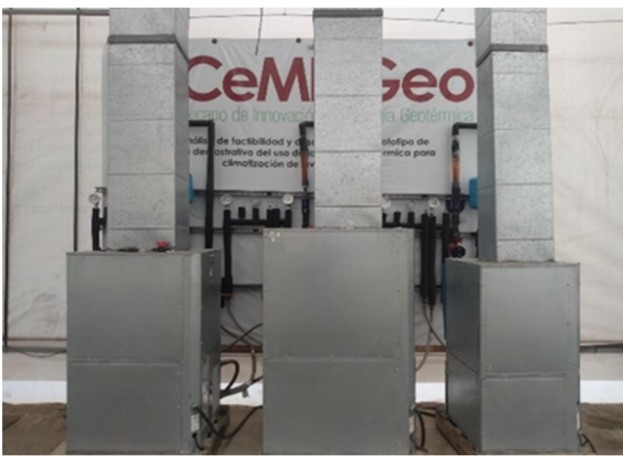

**Figure 7.** Geothermal heat pump systems were installed at the study site.

## 3. Mathematical Model

The mathematical model that describes the performance of the linear GHE used in this study was based on the programming found in the buried pipeline simulation module of the TRNSyS 2017 library. The simulation module consists of a subroutine model of buried pipes in 3 finite dimensions that calculate the dissipation or absorption of heat in the subsoil with water as the working fluid.

The liquid in the tube was modeled as an axial series of isothermal liquid nodes. The mass of the liquid nodes was taken into account in the model, but the assumption was made that the pipe wall material and insulation were massless. Similarly, the assumption was made that the pipe was surrounded by soil whose thermal conductivity, density, and specific heat were known. Figure 8 shows the diagram of radial and axial nodes of the subsoil surrounding the pipe.

The volume of cylindrical soil surrounding the tube is called the near field. The temperature of the nodes in the near field is affected by the energy transferred with the pipe. The near field is, in turn, surrounded by the far field, which is assumed to be a sink or source of infinite energy. In other words, the energy transfer with the far field does not result in a temperature change in the far field. Temperatures in the far field are governed only by depth and time of year. The energy balance at any node in the soil is shown in Equation (1).

$$mCp\frac{dT}{dt} = \dot{Q}_{in} - \dot{Q}_{out} \tag{1}$$

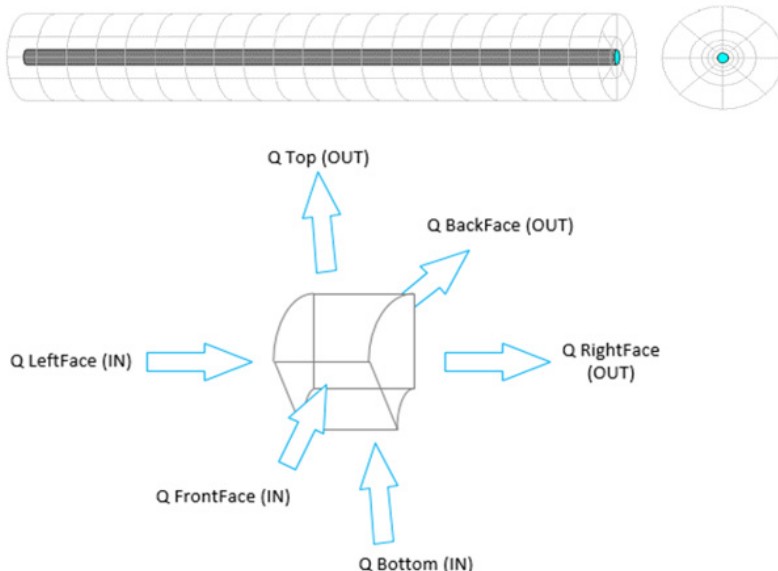

**Figure 8.** Diagram of radial and axial nodes of the subsoil surrounding the pipe.

If a node is contained entirely within the near field and is bounded neither by the far field nor by the pipe wall, its energy balance is made up of purely conductive terms. The equations for $\dot{Q}_{in}$ and $\dot{Q}_{out}$ are:

$$\dot{Q}_{in} = \frac{1}{R_{Bottom}}\left(\underline{T}_{i-1,j,k} - T_{i,j,k}\right) + \frac{1}{R_{LeftFace}}\left(\underline{T}_{i,j-1,k} - T_{i,j,k}\right) + \frac{1}{R_{FrontFace}}\left(\underline{T}_{i,j,k-1} - T_{i,j,k}\right) \tag{2}$$

$$\dot{Q}_{out} = \frac{1}{R_{Top}}\left(T_{i,j,k} + \underline{T}_{i+1,j,k}\right) + \frac{1}{R_{RightFace}}\left(T_{i,j,k} + \underline{T}_{i,j+1,k}\right) + \frac{1}{R_{BackFace}}\left(T_{i,j,k} + \underline{T}_{i+1,j,k}\right) \tag{3}$$

The first term of each equation represents the energy transfer in the radial direction. The second represents the energy transfer in the circumferential direction, and the third represents the energy transfer in the axial direction. The resistors (each of which is written in the form ($\Delta x/kA$) are:

$$R_{Bottom} = \frac{lnln\left(\frac{r_i}{r_{i-1}}\right)}{k_{soil}\left(\frac{2\pi\left(\frac{L_{pipe}}{n_{axial}}\right)}{n_{circ}}\right)} \tag{4}$$

$$R_{Bottom} = \frac{lnln\left(\frac{r_{i+1}}{r_i}\right)}{k_{soil}\left(\frac{2\pi\left(\frac{L_{pipe}}{n_{axial}}\right)}{n_{circ}}\right)} \tag{5}$$

$$R_{LeftFace} = \frac{\left(\frac{L_{pipe}}{n_{axial}}\right)}{k_{soil}\left(\frac{\pi(r_i^2 - r_{i-1}^2)}{n_{circ}}\right)} \tag{6}$$

$$R_{RightFace} = \frac{\left(\frac{L_{pipe}}{n_{axial}}\right)}{k_{soil}\left(\frac{\pi(r_{i+1}^2 - r_i^2)}{n_{circ}}\right)} \tag{7}$$

$$R_{FrontFace} = \frac{\left(\frac{2\pi\, r_{avg}}{n_{circ}}\right)}{k_{soil}\left(\frac{L(r_i - r_{i-1})}{n_{axial}}\right)} \tag{8}$$

$$R_{BackFace} = \frac{\left(\frac{2\pi\, r_{avg}}{n_{circ}}\right)}{k_{soil}\left(\frac{L(r_{i+1}-r_i)}{n_{axial}}\right)} \tag{9}$$

For a fluid node, there are three basic terms in the energy balance. Energy is transferred in and out of the node due to fluid flow, the energy is transferred due to axial conduction between fluid nodes, and energy is transferred between the fluid and the pipe wall. The energy transfer due to flow takes the form:

$$\dot{Q}_{in} = \dot{m}_{fluid}Cp_{fluid}\left(T_{fluid,n} - T_{fluid,n-1}\right) \tag{10}$$

$$\dot{Q}_{out} = \dot{m}_{fluid}Cp_{fluid}\left(T_{fluid,n+1} - T_{fluid,n}\right) \tag{11}$$

The energy flux due to axial conduction takes the form:

$$\dot{Q}_{in} = \frac{k_{fluid}A_{xs}}{L_{node}}\left(T_{fluid,n} - T_{fluid,n-1}\right) \tag{12}$$

$$\dot{Q}_{out} = \frac{k_{fluid}A_{xs}}{L_{node}}\left(T_{fluid,n+1} - T_{fluid,n}\right) \tag{13}$$

The flow of energy between the fluid and the wall depends on the mass flow rate of the fluid in the pipe. The Reynolds and Prandtl numbers are calculated as:

$$Re_{inside} = \frac{4\, \dot{m}_{fluid}}{\pi\, d_{inside,pipe}\, \mu_{fluid}} \tag{14}$$

$$Pr_{inside} = \frac{Cp_{fluid}\, \mu_{fluid}}{k_{fluid}} \tag{15}$$

The Nusselt number is then calculated as:

$$Nu_{inside} = \left(3.66^3 + 1.61^3 Re_{inside}Pr_{inside}\frac{d_{inside,pipe}}{L_{pipe}}\right)^{\frac{1}{3}} \tag{16}$$

for $Re < 2300$ and $\frac{L_{pipe}}{d_{inside,pipe}} > 0.0425 Re_{inside}Pr_{inside}$

$$Nu_{inside} = 4.364 \tag{17}$$

for $Re < 2300$ and $\frac{L_{pipe}}{d_{inside,pipe}} > 0.0425 Re_{inside}Pr_{inside}$

$$Nu_{inside} = 0.0212\left(Re_{inside}^{0.08} - 100\right)Pr_{inside}^{0.4} \tag{18}$$

for $Re > 2300$ and $Pr_{inside} \leq 1.5$

$$Nu_{inside} = 0.012\left(Re_{inside}^{0.87} - 280\right)Pr_{inside}^{0.4} \tag{19}$$

for $Re > 2300$ and $Pr_{inside} > 1.5$

The internal convection coefficient is as follows:

$$h_{inside} = \frac{Nu_{inside}k_{fluid}}{d_{inside,pipe}} \tag{20}$$

The general equation of the energy transferred between the fluid node and the pipe wall node can finally be written as:

$$\dot{Q} = \frac{1}{R_{fluid} + R_{wall} + R_{insul}} \left( T_{boundary} - T_{fluid,n} \right) \qquad (21)$$

where

$$R_{fluid} = \frac{1}{h_{inside} SA_{inside}} \qquad (22)$$

$$R_{wall} = \frac{lnln \left( \frac{r_{pipe,inside} + \frac{r_{pipe,outside} - r_{pipe,inside}}{2}}{r_{pipe,inside}} \right)}{2\pi \, L_{node} \, k_{pipe}} \qquad (23)$$

$$R_{insul} = \frac{lnln \left( \frac{r_{pipe,outside} + \frac{r_{insul,outside} - r_{insul,inside}}{2}}{r_{pipe,outside}} \right)}{2\pi \, L_{node} \, k_{insul}} \qquad (24)$$

Finally, the efficiency in the cooling mode of the GHP is calculated with the following equation:

$$EER = \frac{\dot{Q}}{\sum W_{in}} = \frac{\dot{m}_{fluid} Cp_{fluid} \left( T_{fluid,out} - T_{fluid,in} \right)}{V_{compresor} * I_{compresor} + V_{pump} * I_{pump}} \qquad (25)$$

## 4. Results

### 4.1. Simulator Validation

The validation of the proposed GHE simulator was validated taking as reference the authors Incropera et al. [14] and according to the methodology proposed in this study. Figure 9 shows the results of the comparison between the data obtained by the simulation in TRNSyS 2017 according to the configuration shown previously in Figure 1, compared with the data reported by the reference authors under the same operating conditions. Outlet temperature values are displayed as a function of the length of the reference GHEs, which range from 10 m to 50 m.

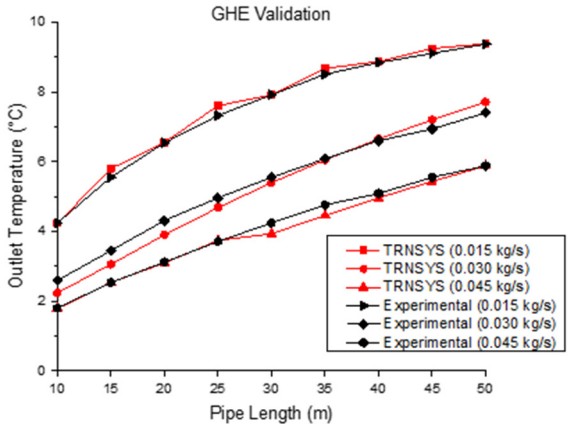

**Figure 9.** Comparison of the pipe length effect in the outlet temperature of the simulator and experimental results.

Likewise, Table 2 shows the deviation between the data obtained in the simulator and the reference information in the three mass flows proposed in the methodology. In this study, it was possible to observe a deviation range that oscillated between 0 and 9%, so it is evident that the results of the simulator have a representative behavior of the phenomenon. For this reason, this simulator is considered a useful tool for linear pipeline GHE simulation.

**Table 2.** The temperature of the fluid at different depths was used in the validation of the GHP simulator results.

| Temperatures Along the GHE | Temperature (°C) | | | | | | | | |
|---|---|---|---|---|---|---|---|---|---|
| | Incropera | | | TRNSYS | | | Deviation % | | |
| | 0.015 kg/s | 0.030 kg/s | 0.045 kg/s | 0.015 kg/s | 0.030 kg/s | 0.045 kg/s | 0.015 kg/s | 0.030 kg/s | 0.045 kg/s |
| T(10 m) | 4.24 | 2.59 | 1.80 | 4.23 | 2.36 | 1.79 | 0% | 9% | 1% |
| T(15 m) | 5.55 | 3.45 | 2.53 | 5.80 | 3.22 | 2.53 | −5% | 7% | 0% |
| T(20 m) | 6.54 | 4.31 | 3.12 | 6.55 | 3.90 | 3.09 | 0% | 9% | 1% |
| T(25 m) | 7.33 | 4.96 | 3.71 | 7.60 | 4.68 | 3.74 | −4% | 6% | −1% |
| T(30 m) | 7.92 | 5.55 | 4.24 | 7.92 | 5.40 | 3.93 | 0% | 3% | 7% |
| T(35 m) | 8.51 | 6.08 | 4.76 | 8.67 | 6.05 | 4.46 | −2% | 0% | 6% |
| T(40 m) | 8.84 | 6.60 | 5.09 | 8.87 | 6.65 | 4.96 | 0% | −1% | 3% |
| T(45 m) | 9.11 | 6.94 | 5.55 | 9.24 | 7.20 | 5.43 | −1% | −4% | 2% |
| T(50 m) | 9.37 | 7.40 | 5.88 | 9.38 | 7.71 | 5.88 | 0% | −4% | 0% |

*4.2. Experimental Validation*

Once the simulator was validated in TRNSyS, experimental runs were carried out according to the operating ranges of the hydraulic pumping system (1, 2, and 3 GPMs). The experimentation consisted of reading the temperatures of the working fluid at the outlet of the GHE when the GHP was in stable operation in refrigeration mode. Simultaneously, the performance coefficient of GHP was obtained in the same period of time. Additionally, the validation of the simulator with the experimental data was carried out to evaluate the deviation that existed between the experimental data of a slinky-type GHE and the linear pipe GHE simulator. Tables 3–5 show the deviation of the simulated data.

**Table 3.** Results and deviation of the performance of the simulation and experimental data at 1 GPM.

| Hour | Tout (Experimental) | Tout (TRNSYS) | Deviation (%) | EER (Experimental) | EER (TRNSYS) | Deviation (%) |
|---|---|---|---|---|---|---|
| 9:15 | 30 | 30.03 | 0.11 | 3.24 | 3.23 | 0.17 |
| 10:00 | 30 | 30.21 | 0.71 | 3.06 | 3.02 | 1.13 |
| 11:00 | 30 | 30.38 | 1.25 | 3.06 | 3.00 | 1.98 |
| 12:00 | 30 | 30.47 | 1.56 | 3.06 | 2.98 | 2.47 |
| 13:00 | 30 | 30.57 | 1.89 | 3.06 | 2.97 | 2.99 |
| 14:00 | 30 | 30.65 | 2.17 | 3.06 | 2.95 | 3.42 |
| 15:00 | 30 | 30.72 | 2.40 | 3.06 | 2.94 | 3.79 |
| 16:00 | 30 | 30.78 | 2.60 | 3.06 | 2.93 | 4.10 |
| 17:00 | 30 | 30.83 | 2.78 | 3.06 | 2.92 | 4.38 |
| 18:00 | 30 | 30.88 | 2.93 | 3.06 | 2.92 | 4.63 |

**Table 4.** Results and deviation of the performance of the simulation and experimental data at 2 GPM.

| Hour | Tout (Experimental) | Tout (TRNSYS) | Deviation (%) | EER (Experimental) | EER (TRNSYS) | Deviation (%) |
|---|---|---|---|---|---|---|
| 9:15 | 30 | 30.04 | 0.15 | 1.74 | 1.74 | 0.19 |
| 10:00 | 30 | 30.06 | 0.21 | 1.74 | 1.74 | 0.26 |
| 11:00 | 30 | 30.08 | 0.27 | 1.74 | 1.73 | 0.34 |
| 12:00 | 30 | 30.10 | 0.34 | 1.74 | 1.73 | 0.42 |

**Table 4.** *Cont.*

| Hour | Tout (Experimental) | Tout (TRNSYS) | Deviation (%) | EER (Experimental) | EER (TRNSYS) | Deviation (%) |
|------|------|------|------|------|------|------|
| 13:00 | 30 | 30.12 | 0.39 | 1.74 | 1.73 | 0.49 |
| 14:00 | 30 | 30.13 | 0.44 | 1.74 | 1.73 | 0.56 |
| 15:00 | 30 | 30.15 | 0.49 | 1.74 | 1.73 | 0.61 |
| 16:00 | 30 | 30.16 | 0.54 | 1.74 | 1.73 | 0.67 |
| 17:00 | 30 | 30.17 | 0.58 | 1.74 | 1.73 | 0.72 |
| 18:00 | 30 | 30.18 | 0.62 | 1.74 | 1.73 | 0.77 |

**Table 5.** Results and deviation of the performance of the simulation and experimental data at 3 GPM.

| Hour | Tout (Experimental) | Tout (TRNSYS) | Deviation (%) | EER (Experimental) | EER (TRNSYS) | Deviation (%) |
|------|------|------|------|------|------|------|
| 9:15 | 30 | 30.07 | 0.25 | 3.09 | 3.07 | 0.67 |
| 10:00 | 30 | 30.42 | 1.39 | 2.99 | 2.87 | 3.80 |
| 11:00 | 30 | 30.66 | 2.19 | 3.05 | 2.87 | 5.98 |
| 12:00 | 30 | 30.81 | 2.70 | 3.05 | 2.83 | 7.37 |
| 13:00 | 30 | 30.90 | 2.99 | 3.05 | 2.80 | 8.16 |
| 14:00 | 30 | 30.99 | 3.30 | 3.05 | 2.78 | 9.01 |
| 15:00 | 30 | 31.07 | 3.56 | 3.05 | 2.75 | 9.70 |
| 16:00 | 30 | 31.13 | 3.77 | 3.05 | 2.74 | 10.29 |
| 17:00 | 30 | 31.19 | 3.96 | 3.05 | 2.72 | 10.79 |
| 18:00 | 30 | 31.24 | 4.12 | 3.05 | 2.71 | 11.23 |

Figures 10–12 show the comparisons of the results of the outlet temperature of the GHE and the EER in the three flows proposed in the experimentation.

The average deviation of the simulator for the GHE outlet temperature corresponds to a value of 2.77% and 3.7% for the EER of the GHP. The average percentage error of the GHE outlet temperature with a flow of 1, 2, and 3 GPMs corresponds to 0.4%, 1.84%, and 2.82%, respectively. On the other hand, the deviation for the EER with a flow of 1, 2, and 3 GPMs corresponds to 0.5%, 2.9%, and 7.7%, respectively.

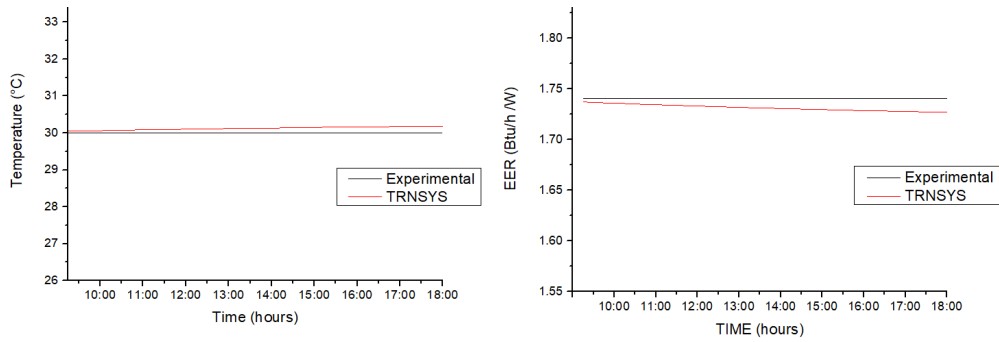

**Figure 10.** Comparison between simulated and experimental results of temperature and EER at a flow rate of 1GPM.

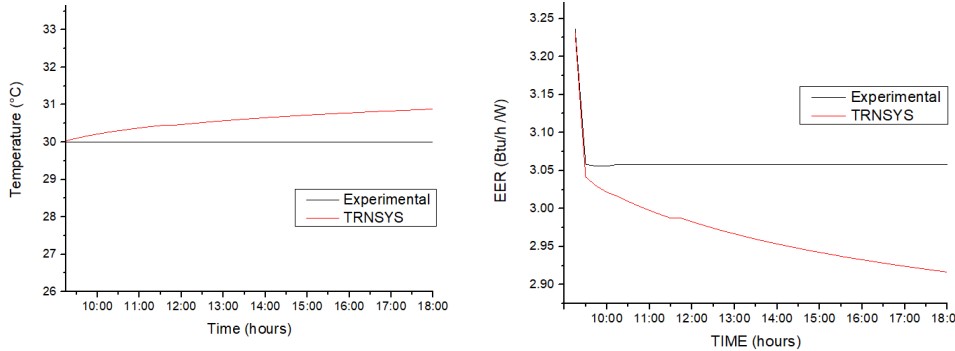

**Figure 11.** Comparison between simulated and experimental results of temperature and EER at a flow rate of 2GPM.

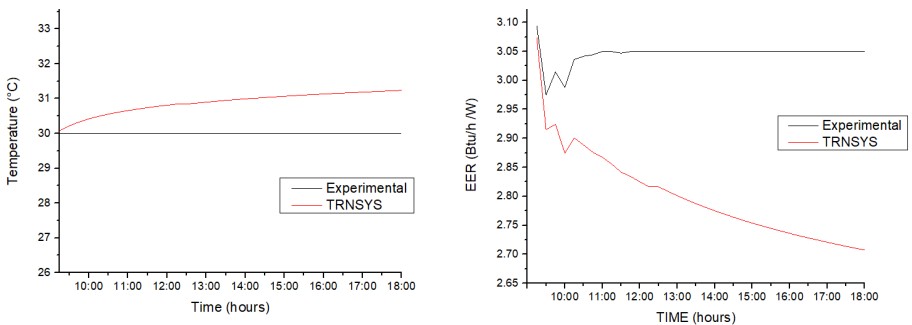

**Figure 12.** Comparison between simulated and experimental results of temperature and EER at a flow rate of 3GPM.

### 4.3. Temperatures and Electrical Consumption

Continuous experimentation was carried out during the year 2020 in the experimental greenhouse, taking the readings of the internal temperature of the greenhouse, the temperature of the subsoil, and the thermal power in heating and cooling mode, as well as the electrical consumption of the GHP. This study was carried out with the purpose of evaluating if the installed capacity of the GHP in the experimental greenhouse was sufficient to satisfy the thermal demand of the enclosure considering the thermal inertia of the subsoil.

Figure 13 shows the results of the measurement of the thermal power absorbed from the subsoil by the GHP in heating mode. In this figure, it can be seen that the critical hours to satisfy the thermal demand are from 5:00 to 7:00, which is when the GHP is operating at its maximum capacity.

Similarly, an analysis of the thermal power dissipated into the subsoil was carried out when the GHP was in cooling mode. This information is found in Figure 14. In this figure, it can be seen that the critical hours to satisfy the thermal demand of the enclosure are from 13:00 to 15:00, which is when the GHP is operating at its maximum capacity. The cooling mode begins its predominance in the month of April and reaches its maximum capacity in the month of July. The heat dissipated in the subsoil in July is 3.4 times higher than in April.

On the other hand, an experiment was carried out to evaluate the electrical consumption of the GHP in each of the modes of operation, which are heating and cooling. This information is found in Figure 15. In this figure, it is shown that the months of April and October are the transition months, which are the months where the predominance of one of the modes of operation changes. Similarly, the figure shows how the months of July and December are the months with the highest electricity consumption for cooling and heating, respectively. The maximum electrical consumption in cooling mode is 11.74% higher compared to the maximum electrical consumption in heating mode.

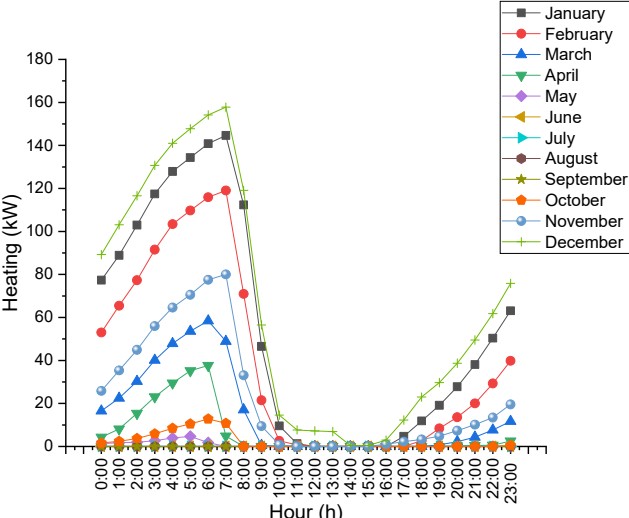

**Figure 13.** The thermal power of the GHP operated in heating mode for a year in Mexicali, Mexico.

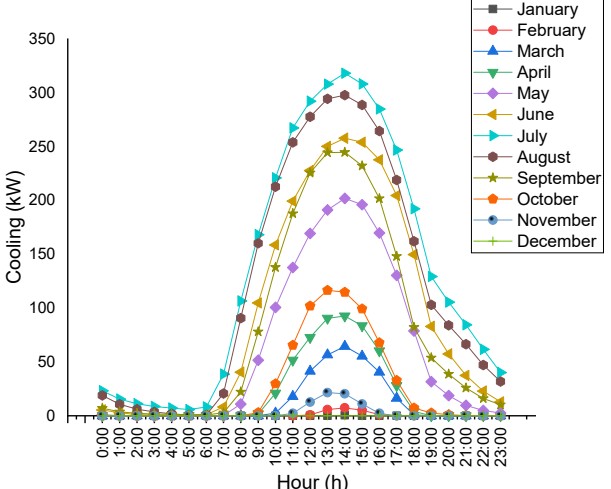

**Figure 14.** The thermal power of the GHP operated in cooling mode for a year in Mexicali, Mexico.

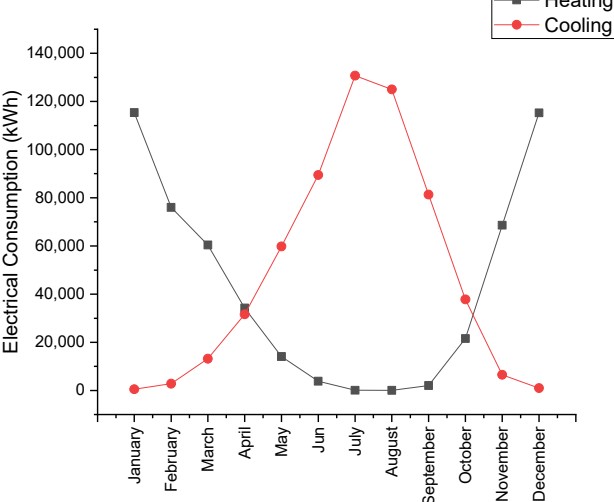

**Figure 15.** Power consumption of the GHP in both modes of operation.

Figure 16 shows the analysis of the daily behavior of temperatures inside the greenhouse air-conditioned with the GHP. This figure shows the effect of GHP air conditioning

by not allowing the greenhouse temperature to exceed 30 °C or fall below 15 °C. The month of March experienced a greater variation in the internal temperature of the greenhouse since its extreme temperatures were within the operating temperature range of the GHP. This variation represents a temperature increase of 45.8% from 06:00 to 14:00.

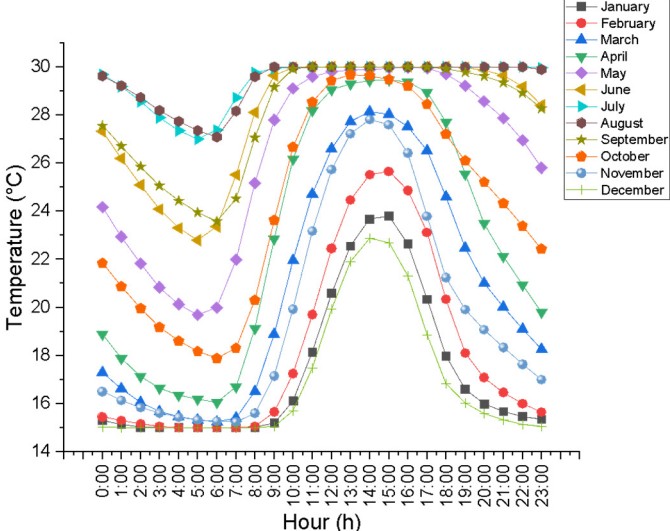

**Figure 16.** Daily behavior of temperatures inside the greenhouse heated with GHP.

Similarly, a study of the daily behavior of subsoil temperatures in the area where the GHEs are installed was carried out. This information is shown in Figure 17. This figure shows how the subsoil is saturated in the months of July and August. The temperature difference from the beginning of the cooling season (March) to the peak of the season (August) represents an increase of 42.18%. On the other hand, the temperature difference from the beginning of the heating season (October) to the minimum point of the season (February) represents a decrease of 52.22%.

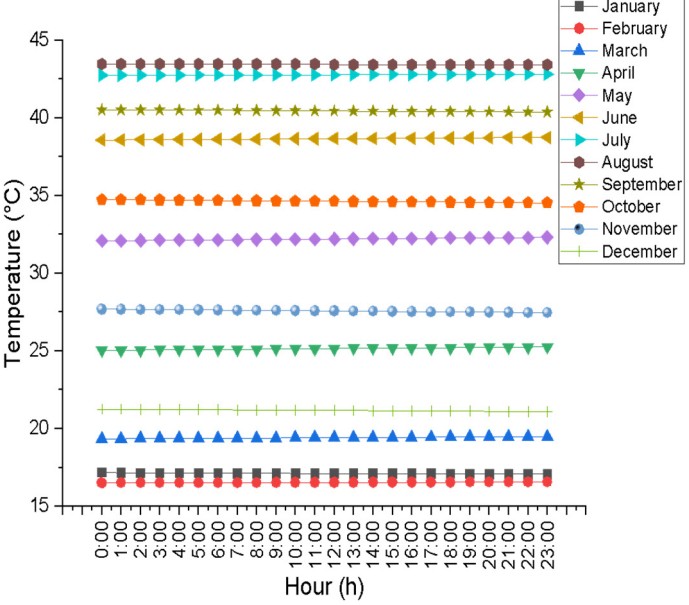

**Figure 17.** Daily behavior of subsoil temperatures at different months.

## 5. Conclusions

In this study, the results of a simulation supported by experimental data to analyze the performance of a greenhouse operating with a geothermal heat pump located in Mexico in an arid zone were presented. The study was conducted during a complete year to include the heating and cooling mode performance while tomatoes were harvested. The results show that the average deviation of the simulator for the geothermal heat exchanger (GHE) outlet temperature corresponded to a value of 2.77% and 3.7% for the EER of the GHP. The average error percentage of the GHE outlet temperature with a flow of 1, 2, and 3 GPM corresponded to 0.4%, 1.84%, and 2.82%, respectively. The deviation for the EER with a flow of 1, 2, and 3 GPM corresponded to 0.5%, 2.9%, and 7.7%, respectively. According to the results, the critical hours in the heating mode were in the range from 5:00 to 7:00. The critical hours in the cooling mode were in the range of 13:00 to 15:00. The months of April and October are the transition months, that is, the months in which the predominance of one of the modes of operation changes. The months of July and December are the months with the highest electricity consumption for cooling and heating, respectively. The installed capacity of GHP in the experimental greenhouse meets the thermal demand of the enclosure. However, it is recommended that for areas similar to those in the study, the number of hours of operation be reduced in the last 2 weeks of July and the first 2 weeks of August due to subsoil heat saturation.

**Author Contributions:** Conceptualization, J.O.R.V. and J.A.Q.L.; methodology, A.A.R. and J.A.S.M.; software, A.H.R.P. and J.A.C.S.; validation, A.H.R.P. and J.A.C.S.; formal analysis, J.O.R.V., A.A.R., J.A.Q.L. and A.M.H.; investigation, J.O.R.V. and J.A.Q.L.; resources, A.M.H.; data curation, J.O.R.V., J.A.Q.L. and A.M.H.; writing—original draft preparation, J.O.R.V.; writing—review and editing, J.A.Q.L.; visualization, P.F.R.E. and J.A.S.M.; supervision, F.L.C.; project administration, A.M.H.; funding acquisition, A.A.R. and A.M.H. All authors have read and agreed to the published version of the manuscript.

**Funding:** The experimental greenhouse project was financed by the Mexican Center for Innovation in Geothermal Energy CEMIEGeo in the P10 project called "Feasibility analysis and development of a prototype demonstration project for the use of geothermal energy for air conditioning greenhouses".

**Institutional Review Board Statement:** Not applicable.

**Informed Consent Statement:** Not applicable.

**Data Availability Statement:** Not applicable.

**Acknowledgments:** This article was developed thanks to the collaboration agreement between the Autonomous University of Baja California (UABC) and the Polytechnic University of Baja California (UPBC). UPBC provided facilities for the development of experimentation and dissemination of this research. Similarly, the UABC supported the development of the simulator shown in this article. We thank these institutions for making the development of this research possible. All individuals included in this section have consented to the acknowledgement.

**Conflicts of Interest:** The authors declare no conflict of interest.

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
