# Peer review of "Theoretical-Experimental Analysis of the Performance of Geothermal Heat Pumps for Air Conditioning Greenhouses in Arid Zones"

_processes, doi:10.3390/pr10091682_

Round 1

Reviewer 1 Report

The authors used experimental and simulation methods to analyze the performance of geothermal heat pumps. I think this paper has not been well organized. It is difficult to read. Even though the author mentioned the current study is a research opportunity, I could not judge the novelty from the introduction. The main points I am concerned about are as follows for reference in the revision.

  1. The introduction should not be a simple list of relevant studies. You should summarize and classify the previous studies and point out the missing part in the literature.
  2. Why did you separate section 2.2.1 from section 3?
  3. What do the symbols mean in Fig.4 and 5, such as Type952 and Type 647?
  4. In Table 2-5: why did you use the term uncertainty? It is more like a predicting deviation.
  5. Please explain why the deviation increased with hours.
  6. What is the atmospherical temperature outside the greenhouse at different months?
  7. There is a gap between the experimental work(section 4.3) and simulation work (sections 4.1 and 4.2). Even though they are all related to the geothermal heat pumps, I feel like they are forced together. The experimental work(section 4.3) has nothing related to simulation work. The authors should explain more why these two studies are interconnected.

Reviewer 2 Report

Author conducted a good research on theoretical-experimental analysis of the performance of geo-thermal heat pumps for air conditioning greenhouses in arid. There are some suggestions authors need to accept:

1. Innovation should be further emphasized in the introduction

2. Many parameters in this paper do not give specific values.

3. Many formulas in this paper are unclear in origin and derivation process.

4. The formula font is too small.

5. The conclusion is not so good.

6. The depth and breadth of literature analysis in the introduction are insufficient. Theoretical-experimental analysis of other system can be cited and studied, such as "Research on the vibration model and vibration performance of cold orbital forging machines". This study conduct theoretical-experimental analysis of manufacturing system to analyze the manufacturing process and proposed the methods to improve the quality of the production.

I recommend that author revised the paper according to the above advises.

Round 2

Reviewer 1 Report

The authors have already well addressed my questions. I can recommend this manuscript be published now.

Reviewer 2 Report

This paper can be accepted.